# Heatwaves and dengue outbreaks in Hanoi, Vietnam: New evidence on early warning

**Jian Cheng**[1], **Hilary Bambrick**[1], **Laith Yakob**[2], **Gregor Devine**[3], **Francesca D. Frentiu**[4], **Do Thi Thanh Toan**[5], **Pham Quang Thai**[5,6], **Zhiwei Xu**[1], **Wenbiao Hu**[1] *

**1** School of Public Health and Social Work, Institute of Health and Biomedical Innovation, Queensland University of Technology, Brisbane, Australia, **2** Department of Disease Control, London School of Hygiene and Tropical Medicine, London, United Kingdom, **3** Mosquito Control Laboratory, QIMR Berghofer Medical Research Institute, Brisbane, Australia, **4** School of Biomedical Sciences, Institute of Health and Biomedical Innovation, Queensland University of Technology, Brisbane, Australia, **5** Institute of Preventive Medicine and Public Health, Hanoi Medical University, Hanoi, Vietnam, **6** Communicable Disease Control Department, National Institute of Hygiene and Epidemiology, Hanoi, Vietnam

* w2.hu@qut.edu.au

## Abstract

### Background

Many studies have shown associations between rising temperatures, El Niño events and dengue incidence, but the effect of sustained periods of extreme high temperatures (i.e., heatwaves) on dengue outbreaks has not yet been investigated. This study aimed to compare the short-term temperature-dengue associations during different dengue outbreak periods, estimate the dengue cases attributable to temperature, and ascertain if there was an association between heatwaves and dengue outbreaks in Hanoi, Vietnam.

### Methodology/Principal findings

Dengue outbreaks were assigned to one of three categories (small, medium and large) based on the 50th, 75th, and 90th percentiles of distribution of weekly dengue cases during 2008–2016. Using a generalised linear regression model with a negative binomial link that controlled for temporal trends, temperature variation, rainfall and population size over time, we examined and compared associations between weekly average temperature and weekly dengue incidence for different outbreak categories. The same model using weeks with or without heatwaves as binary variables was applied to examine the potential effects of extreme heatwaves, defined as seven or more days with temperatures above the 95th percentile of daily temperature distribution during the study period. This study included 55,801 dengue cases, with an average of 119 (range: 0 to 1454) cases per week. The exposure-response relationship between temperature and dengue risk was non-linear and differed with dengue category. After considering the delayed effects of temperature (one week lag), we estimated that 4.6%, 11.6%, and 21.9% of incident cases during small, medium, and large outbreaks were attributable to temperature. We found evidence of an association between heatwaves and dengue outbreaks, with longer delayed effects on large outbreaks (around 14 weeks later) than small and medium outbreaks (4 to 9 weeks later). Compared with non-heatwave years, dengue outbreaks (i.e., small, moderate and large outbreaks

**Data Availability Statement:** Climate data are publicly available for researchers at National Oceanic and Atmospheric Administration website (https://www.noaa.gov/); Dengue data are available

from the National Institute of Hygiene and Epidemiology, Vietnam upon request for researchers via e-mail (phamquangthai@gmail.com) who meet the criteria for accessing confidential data since public deposition would breach compliance with the protocol approved by the ethics board.

**Funding:** The work was supported by National Health and Medical Research Council (NHMRC) of Australia, project grant APP 1138622. The funder had no role in study design, data collection and analysis, decision to publish, or preparation of the manuscript.

**Competing interests:** The authors have declared that no competing interests exist.

combined) in heatwave years had higher weekly number of dengue cases ($p<0.05$). Findings were robust under different sensitivity analyses.

## Conclusions

The short-term association between temperature and dengue risk varied by the level of outbreaks and temperature seems more likely affect large outbreaks. Moreover, heatwaves may delay the timing and increase the magnitude of dengue outbreaks.

### Author summary

Dengue fever is one of the most common mosquito-borne viral diseases. Weather extremes such as El Niño event and extreme hot summer can affect dengue incidence rate and dengue outbreaks. More frequent, more intensive and longer lasting heatwaves in the 21st century is anticipated because of global warming, making it necessary to investigate the association between heatwaves and dengue outbreaks. In this study, we estimated 4.6%, 11.6%, and 21.9% of incident dengue cases during small, medium, and large outbreaks attributable to temperature in Hanoi, Vietnam. We also found evidence of an association between heatwaves and dengue outbreaks, with longer delayed effects on large outbreaks than small and medium outbreaks. Compared with non-heatwave years, dengue outbreaks in heatwave years had higher number of dengue cases. Heatwave weather may represent an emerging risk factor or predicator of dengue outbreaks in tropical regions. Future dengue prediction models incorporating heatwaves may help increase the accuracy of predictability.

## Introduction

As the most common mosquito-borne viral disease of public health significance, dengue affects more than 126 countries and costs billions of US dollars each year [1, 2]. Vector control remains the most viable approach to curb the transmission of dengue. However, vector control measures seem unable to stem the rising number of arbovirus epidemics and the spatial spread of endemic transmission [1, 3]. The underlying drivers of dengue transmission have been investigated previously and include increased air travel and trade and urban crowding [4, 5]. The impacts of weather conditions on dengue occurrence is attracting extensive interest among the research community and policy makers, especially in the context of climate change [6–9].

Increasing epidemiological and experimental evidence points to a link between outdoor temperature and dengue incidence [6, 10–14]. The exposure-response association between temperature and dengue incidence is characterised as a bell shape, with the risk of dengue peaking at an optimal temperature, also referred to as the inflection point [11, 14]. Dengue incidence has apparent seasonal pattern, but previous studies looking at short-term temperature effects on dengue incidence mainly included all dengue cases during study period or peaking period [11, 14], making it hard to know the relative contribution of temperature to different sizes of dengue outbreaks. Because dengue outbreaks within a region can vary in terms of the timing, magnitude and duration, it is necessary to examine temperature-dengue associations during different outbreak periods and estimate temperature-related dengue burden.

High temperatures during heatwaves can suppress the replication of virus and activity among mosquitoes, as well as alter human behaviours that are not favourable to transmission

or spread of dengue. For instance, a recent study suggested that during a heatwave, high temperature actually had an inhibitive effect on mosquito abundance [15]. However, there is evidence that the epidemics or outbreaks of some vector-borne diseases such as West Nile fever could happen several weeks or months after heatwaves [16–18]. Some studies also reported an potential association of weather extremes such as El Niño event and extreme hot summer with higher dengue incidence rate and dengue outbreaks [9, 19, 20]. The available evidence indicates that there might be an association between heatwaves and dengue occurrence. A heatwave is referred to a sustained period (days to weeks) of extreme high temperature [21], which is different from previously studied weather extremes (El Niño event and extreme hot summer) that describe a weather condition within a longer time period such as the whole year and the whole summer [9, 19, 20]. As a result of climate change of 21st century, heatwaves will become more frequent, more intensive and longer-lasting, making it urgent to investigate the impact of heatwaves on dengue outbreaks [7, 8].

In this study, we collected the dengue data in Hanoi, Vietnam, a hot spot for dengue transmission in Asia-Pacific area [22]. The aims of this study were to characterize and compare short-term associations between temperature and dengue infection during different outbreak periods; to estimate the extent to which dengue cases can be attributed to temperature; and to ascertain if there is a link between heatwaves and subsequent dengue outbreaks.

## Methods

### Ethics statement

This study was approved by University Human Research Ethics Committee of Queensland University of Technology (1800000058).

### Study location

Hanoi, the capital of Vietnam, features a subtropical climate with plentiful rainfall and has 7.3 million population with a population density of 2,182 persons per $km^2$ in 2016 (https://www.gso.gov.vn/default_en.aspx?tabid=774). Historically, there was an increase of 1.38% per year in the dengue incidence over the period 1999–2008 in Hanoi [23] and local temperature plays a role in affecting dengue incidence [22].

### Data collection

We collected weekly number of dengue cases for all the districts of Hanoi, Vietnam during 2008–2016 from Hanoi Center for Preventive Medicine. There were no missing data for any week over the study period. Daily records of maximum temperature, minimum temperature, and rainfall for Hanoi over the same period were obtained from the National Oceanic and Atmospheric Administration (https://www.noaa.gov/). For the data analysis, we used the weekly mean temperature calculated by averaging the maximum and minimum temperatures within a week as the exposure index to investigate the temperature effects on dengue incidence [6, 14]. There is also evidence that short-term temperature variation can affect dengue virus transmission [10]. Weekly temperature variation was calculated from the standard deviation of daily mean temperature within a week [24].

### Definitions of dengue outbreaks and heatwaves

Although there are many previous studies focusing on dengue outbreaks, the existing definition for dengue outbreaks is mixed [25]. Importantly, different definitions could result in inconsistent findings on the dengue outbreak characteristics in terms of frequency and

duration [25]. To ease the interpretation of effect estimates, we employed an easy-to-under-stand approach to defining dengue outbreaks [12]. Specifically, dengue outbreaks were classified into three levels: small, medium, and large. Small outbreaks were defined as weekly total number of dengue cases above the 50th percentile but below 75th percentile of weekly dengue distribution during the study period, medium outbreaks as weekly total number of dengue cases above the 75th percentile but below 90th percentile of weekly dengue distribution, and large outbreaks as weekly total number of dengue cases above 90th percentile of weekly dengue distribution. Moreover, some other definitions with different thresholds for small (55th and 60th percentiles), medium (70th and 77th percentiles) and large (91st and 92nd percentiles) outbreaks were also used to check the robustness of our findings.

Similarly, there has been no globally accepted "heatwave" definition [26], and we defined a heatwave with the combination of intensity and duration of heat. This method has been used in our previous studies [21, 26]. Specifically, a heatwave was defined as seven or more days with temperature above the 95th percentile of daily temperature distribution during the study period. To avoid the possibility that using one heatwave definition generates significant results by chance, we also tested other heatwave definitions (seven or more days with temperature exceeding 90th percentile of daily temperature distribution; two or more days with tempera-ture exceeding 99th percentile of daily temperature distribution; and three or more days with temperature exceeding 99th percentile of daily temperature distribution) in the sensitivity analysis.

## Statistical methods

Our data analysis consisted of three stages. In the first stage, we fitted the association between weekly mean temperature and weekly total number of dengue cases using a negative binomial generalized linear model combined with a distributed lag non-linear model [6, 27, 28]. Because short-term temperature variation (e.g., diurnal temperature range) and rainfall were found to have impacts on dengue incidence [14, 27], to consider the influence of confounders, we included in the model weekly temperature variation and weekly average rainfall, both using a natural cubic spline smoothing function with three degrees of freedom. To control for the long-term trend, we included "year" as a categorical variable in the model. An auto-regressive term for weekly total number of dengue cases was also incorporated in regression model, because the observed weekly counts are unlikely to be independent. Additionally, each year's population in log scale were also included in the model as an offset to control for potential con-founding effect of demographic shifts over time.

To allow for the non-linear relationship between temperature and dengue risk, we used a natural cubic spline smoothing function for temperature. The degree of freedom for smooth-ing function was determined by the lowest Akaike's information criterion (AIC) value of the model. Considering the incubation period (1–2 weeks) of dengue infection [14, 29], in addi-tion to the first week modelling of temperature-dengue relationship, we also used a lag of one week to track the short-term delayed effect of temperature on dengue occurrence. To measure the size of temperature effects, we chose 18˚C as the reference to calculate the relative risk (RR) and 95% confidence interval (CI) for different temperature points. The reason for this choice was because previous study found that dengue transmission can occur between 18˚C and 34˚C [13]. This analysis (first stage) was repeated for different levels of dengue outbreaks. The goodness-of-fit of model was validated via checking the distribution and auto-correlation of model residuals (S1–S3 Figs).

In the second stage, we estimated the extent to which dengue cases were attributable to tem-perature following the method in our previous work [21, 30]. Relevant estimates were reported

as attributable fraction and attributable number. This analysis was mainly used to compare the relative contribution of temperature to different levels of dengue outbreaks.

In the third stage, we fitted the association between heatwaves and dengue cases using a negative binomial generalized linear model. Similar to our previous analyses [21], heatwaves were included as the binary variable ('1' if a heatwave occurred in a week, otherwise, '0' was specified). Except for the temperature, all the confounders in the first-stage analysis were controlled for here.

To explore if heatwaves affect the magnitude of dengue outbreaks, we compared the weekly number of dengue cases across different outbreak levels in years with and without heatwaves using Wilcoxon signed-rank test. For years without heatwave, all outbreak weeks were selected, whereas for years with heatwave, we only selected outbreak weeks after heatwaves. Considering the delayed effect of heatwaves, we also limited the selection of outbreak weeks to the period several weeks after heatwaves; according to the third-stage analysis, the week(s) at which the delayed effect of heatwaves was significant was selected.

All statistical analyses were conducted in R software (version 3.4.0). The packages "dlnm" and "MASS" were used to construct the negative binomial regression model.

## Results

This study included 55,801 dengue cases, with an average of 119 (range: 0–1,454) cases per week. Of these, 12.7% were from small outbreaks, 22.9% were from medium outbreaks, and 59.9% were from large outbreaks. Fig 1 shows the weekly distributions of dengue cases, temperature, and heatwaves. According to the aforementioned definition of dengue outbreaks, small outbreaks occurred in all years and medium outbreaks in eight out of nine years, except the year 2012. Large dengue outbreaks were observed in five out of nine years, with successive occurrence in the first two and the last two years. The average value of weekly mean temperature was 24.3˚C (range: 10.2–33.1˚C). When defining the heatwave as seven or more days with temperature above the 95th percentile of daily temperature distribution, heatwaves were observed only in 2010 and 2015. Many more heatwaves were seen when defining the heatwave as two or more days with temperature above the 99th percentile of daily temperature distribution.

Fig 2 shows the short-term exposure-response relationship between temperature and dengue risk. Within the first week of temperature change (Fig 2A and 2E), there was a bell shape curve for small and large outbreaks, namely the risk of dengue continually increased up to a temperature threshold (i.e., inflection point) and then started to decrease. No inflection point was observed for medium outbreaks (Fig 2C). The largest effect estimates for small, medium and large outbreaks were seen at 24.1˚C (RR: 1.40; 95%CI: 1.22–1.61), 33.1˚C (RR: 1.83; 95% CI: 1.09–3.08), and 22.6˚C (RR: 1.85; 95%CI: 1.44–2.39), respectively.

After considering the delayed effect of temperature (one week lag), a broadly linear relationship between temperature and dengue risk was observed for medium and large outbreaks (Fig 2D and 2F). The largest effect estimates were at 33.1˚C for both medium (RR: 2.92 95%CI: 1.80–4.75) and large outbreaks (RR: 1.90; 95%CI: 1.31–2.76), which is different from that for small outbreaks at 23.8˚C (RR: 2.26; 95%CI: 1.78–2.86) (Fig 2B).

Regarding other weather variables, the association of dengue risk with rainfall and weekly temperature variation was not observed, except for rainfall increases within the first week that were associated with higher dengue risk during medium outbreaks (S4 and S5 Figs).

Fig 3 presents the estimated attributable components for temperature during different dengue outbreak periods. For large dengue outbreaks (Fig 3A, 3B and 3C), temperature within the first week was responsible for 18.5% of total cases, corresponding to 1,148.1 cases or 16.7 cases

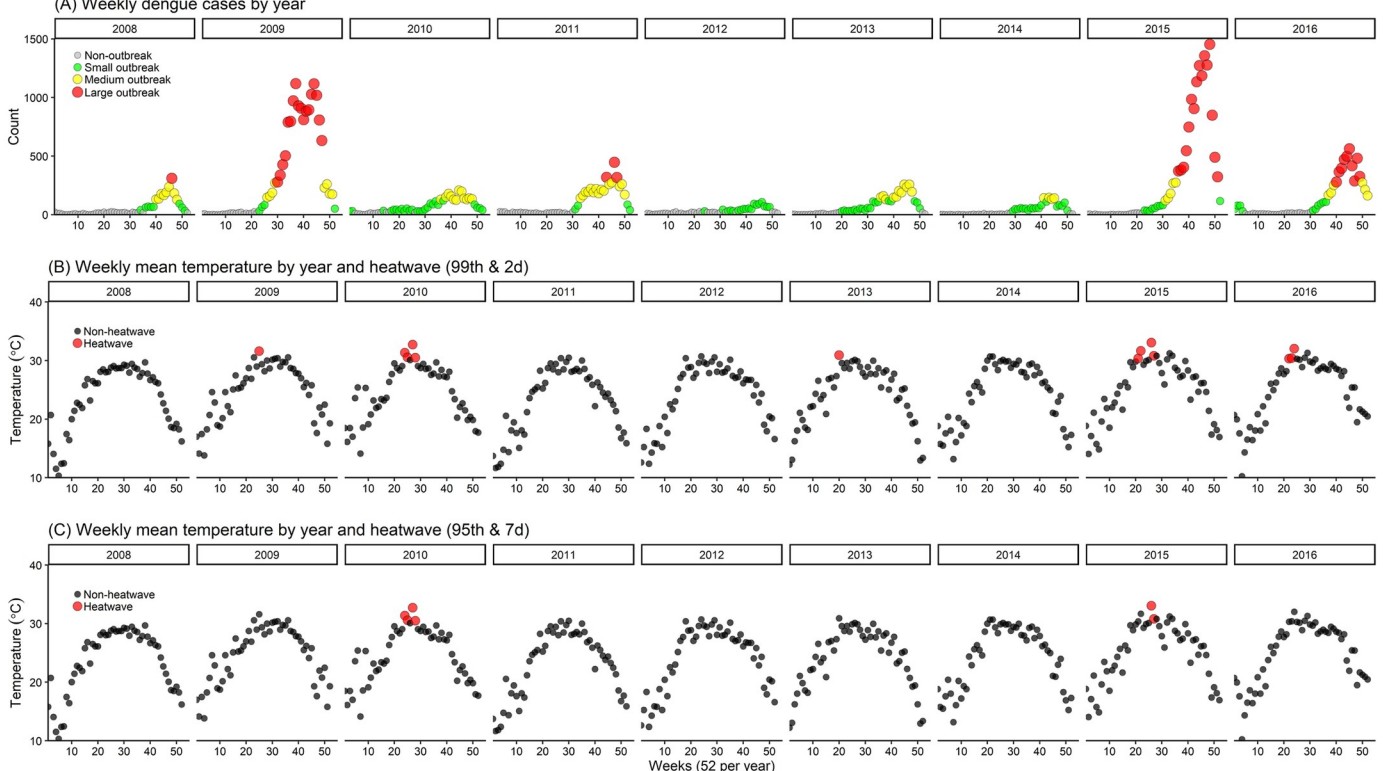

**Fig 1. The weekly distribution of dengue cases, temperature and heatwaves in Hanoi, Vietnam, 2008–2016.** (A) Weekly dengue cases were categorized into non-outbreak (grey points), small outbreak (green points), medium outbreak (yellow points) and large outbreak (red points); (B) and (C) used two heatwave definitions:: two or more days with temperature exceeding 99th percentile of daily temperature distribution (B) and seven or more days with temperature exceeding 95th percentile of daily temperature distribution (C).

per 100,000 people. Similar figures were seen for the temperature impacts one week later (Fig 3D, 3E and 3F). However, much smaller figures were estimated for small and medium outbreaks. For example, only 1.1% of total cases or 70.6 cases or 1 case per 100,000 people were attributable to temperature during small dengue outbreaks.

Fig 4 reports the estimated risk of dengue associated with heatwaves. Significant increases in dengue incidence were observed at lag 7 (RR: 1.47, 95%CI: 1.05–2.05) and lag 9 (RR: 1.51, 95%CI: 1.03–2.19) for small outbreaks, and at lag 7 (RR: 1.41, 95%CI: 1.04–1.91) for medium outbreaks. Much longer delayed effects on large outbreaks were seen at lag 14 (RR: 1.47, 95% CI: 1.08–1.99).

Fig 5 compared the magnitude of dengue outbreaks in years with and without heatwaves. Higher weekly number of dengue cases in heatwave years than in non-heatwave years was observed for small outbreak when considering all outbreak weeks after heatwave ($p<0.05$) (Fig 5A), as well as for small and moderate outbreaks after considering delayed effects of heatwave ($p<0.05$) (Fig 5B). In both cases, heatwave years consistently saw higher weekly number of dengue cases for all outbreaks (i.e., small, moderate and large outbreaks combined) ($p<0.05$).

To check the robustness of our findings, we performed several sensitivity analyses. Different thresholds to define the level of dengue outbreaks and different heatwave definitions were used, which generated similar results (S6 and S7 Figs).

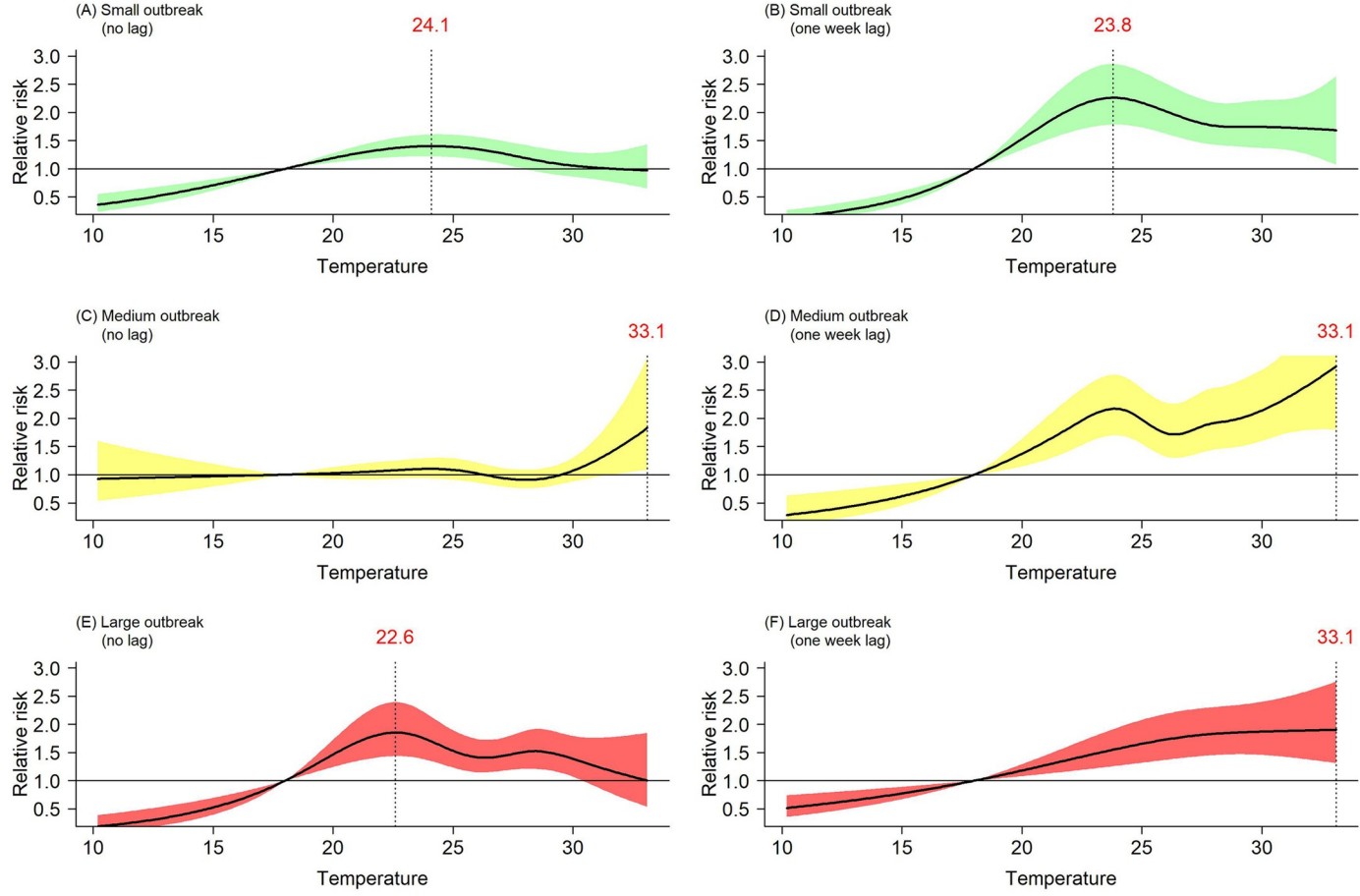

**Fig 2. Exposure-response relationship between temperature and risk of dengue.** (A–F): Black lines indicate the relative risk and the shaded area the 95% confidence interval; dotted lines are the threshold temperature with the highest relative risk. Confounders included in the model in (A–F) are weekly temperature variation, weekly average rainfall, long-term trend, an auto-regressive term, and each year's population in log scale.

## Discussion

This study found that short-term exposure-response relationship between temperature and dengue incidence not only was non-linear but also differed between the categories of outbreak size. Comparatively, temperature had the greatest impacts on large dengue outbreaks, with an estimated 21.9% of total cases or 19.8 cases per 100,000 people attributable to the delayed effects of temperature (one week lag). Moreover, we found a significant association between heatwaves and dengue outbreaks, with longer delayed effects on large outbreaks than small and medium outbreaks. Heatwaves not only delay the timing of dengue outbreaks but also increase its magnitude.

In the past decades, dengue outbreaks have reached many regions of the world and temperature has been asserted as an important driver of dengue transmission [6, 12, 31–33]. Although existing epidemiological and experimental studies have well-documented non-linear relationship between temperature and dengue incidence or transmission [11, 14], little is known about temperature effects on different levels of outbreaks. The present study demonstrated distinct non-linear relationships across the level of dengue outbreaks. Temperature threshold is an important feature of non-linear relationship [14]. For both small and medium outbreaks, thresholds were relative stable, while a larger range was observed for large outbreaks (Fig 2).

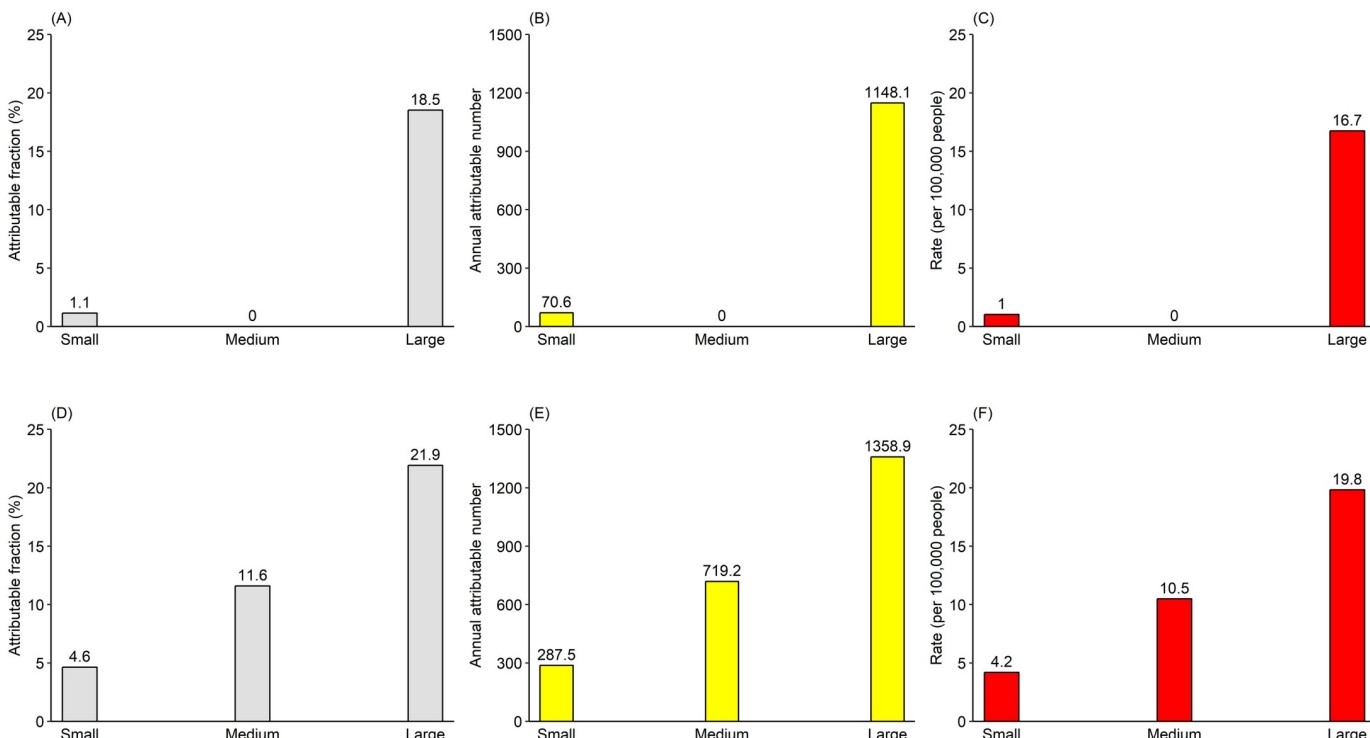

**Fig 3. Estimated attributable dengue cases for temperature, reported as attributable fraction, attributable number, and rate (per 100,000 people).** (A–C) did not consider the lag effects of temperature; (D–F) considered one week lag effects of temperature.

The underlying reasons for this inconsistency are hard to know because the drivers behind dengue transmission are multifaceted (e.g., temperature, relative humidity, population mobility, population of virus-infected mosquitoes, etc.). Nevertheless, the identified temperature thresholds could be used to tailor the local public health measures such as health resource allocation and tiered early warning systems in response to different levels of outbreaks [14].

So far, few studies have assessed temperature-related dengue burden, measured as attributable fraction, attributable number or disability-adjusted life years [30]. Burden of disease assessment considers health-related information such as relative risk and prevalence of exposure among population, which is different from the widely used risk assessment like relative risk or odds ratio that is inadequate to reflect the magnitude of health impacts of temperature [30]. Using a widely used approach in the field of climate change and burden of disease [30, 34], we found that temperature was responsible for a large number of dengue cases, with more temperature-related cases from large outbreaks. In other words, temperature may play an important role in affecting dengue outbreaks, especially the large outbreaks. It thus can be expected that rising temperature in the context of global warming will lead to more dengue cases or outbreaks in the absence of effective countermeasures [6, 12]. It would be beneficial to consider the contribution of temperature for policy-makers when developing dengue burden mitigation strategies—an effective dengue forecasting system based on reliable temperature data would undeniably benefit public health [12]. Hence, more of such efforts are required to assess the association between dengue burden and temperature so as to deepen our understanding of climate change impacts. Cost-effectiveness of developing and implementing early warning system that incorporates the short-term effects of temperature is a pressing future research area.

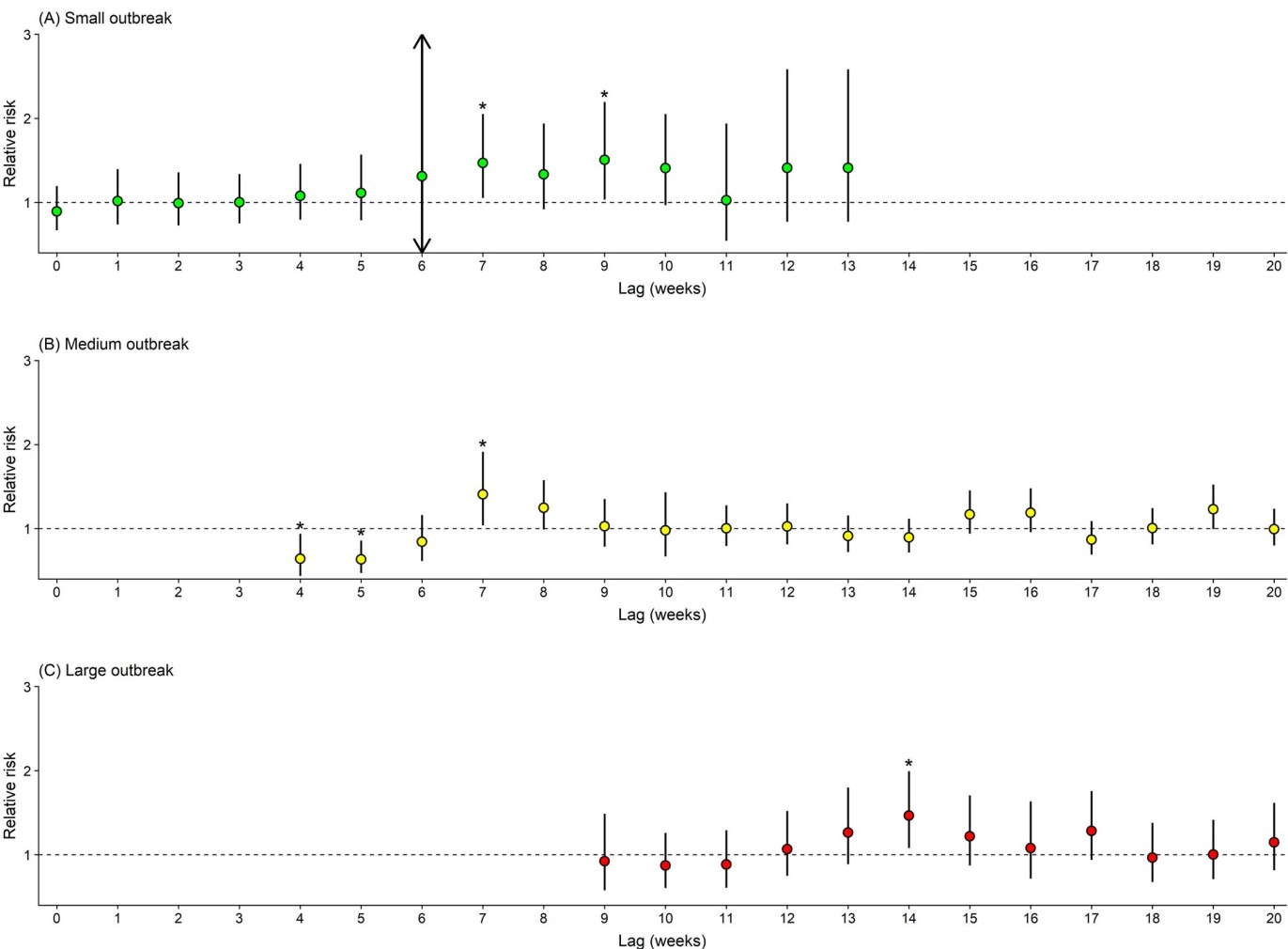

**Fig 4. Estimated relative risk of heatwaves on dengue incidence during different outbreaks.** (A–C): Heatwaves were defined as seven or more days with temperature exceeding 95th percentile of daily temperature distribution.

Heatwave weather is a well-recognised risk factors for health, such as increasing total, cardiovascular and respiratory mortality [21, 30]. There is also growing attention to heatwaves and vector-borne diseases [16–18]. For example, West Nile Virus outbreaks in Romania (1996), New York City (1999) and Israel (2000) were recorded after a heatwave [16, 17]. Researchers also reported an potential association of weather extremes such as extreme hot summer and El Niño events (favouring the occurrence of heatwaves) with higher dengue incidence rate or dengue outbreaks [19, 20, 35]. Here, we observed that large dengue outbreaks in years 2009, 2015, and 2016 occurred several weeks after heatwaves (Fig 1B). The delayed effects of temperature on non-infectious diseases have been well documented in the literature [36, 37]; similarly, it may take some time to observe the effects of heatwaves on dengue outbreaks. A series of several processes link anomalous temperature extremes with outbreaks: following the initially negative effects of heatwaves decreasing local mosquito populations [15], weather conditions occur that are favourable for mosquito growth and reproduction [38], ultimately resulting in an accumulation of dengue infection and outbreaks. The hypothesis for the linkage of heatwaves and subsequent dengue outbreaks was further statistically tested in the present

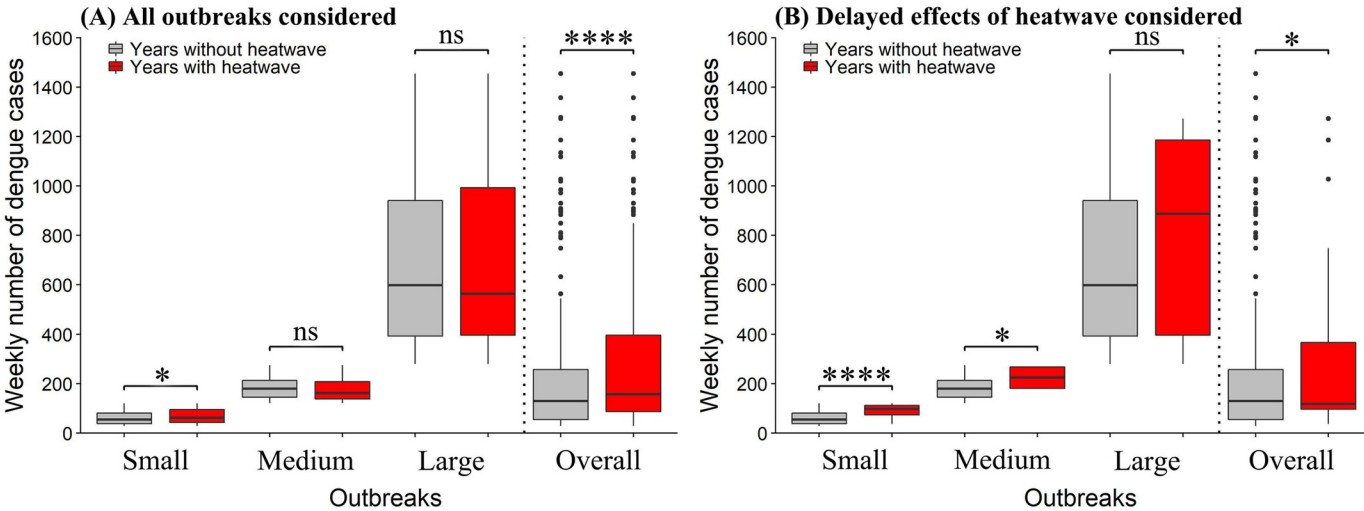

**Fig 5. Comparison of magnitude of small, medium and large outbreaks in years with and without heatwaves.** (A and B): Comparisons were made using four heatwave definitions (seven or more days with temperature exceeding 95th percentile of daily temperature distribution; seven or more days with temperature exceeding 90th percentile of daily temperature distribution; two or more days with temperature exceeding 99th percentile of daily temperature distribution; three or more days with temperature exceeding 99th percentile of daily temperature distribution); (A and B): ns indicates nonsignificant, $p>0.05$; * indicates $p<0.05$; **** indicates $p<0.001$.

study. Interestingly, we found evidence of significant adverse effects of heatwaves (Fig 4). Noticeably, heatwaves had a much longer delayed effect on large dengue outbreaks than small and medium outbreaks, which was also validated by the sensitivity analyses using different heatwave definitions (S7 Fig). Moreover, we observed higher weekly number of dengue cases for all types of outbreaks (i.e., small, moderate and large outbreaks combined) in years with heatwave than in years without heatwave. These findings may suggest that not only have heatwaves delayed the timing of dengue outbreak but also increased the magnitude of outbreak. When heatwaves are predicted to happen, proactive measures should be taken to prevent the upcoming dengue outbreaks or reduce its impacts. Because small and medium heatwaves are more likely to happen in the first few weeks after heatwaves and large dengue outbreaks may happen several months after heatwaves, different levels of early warning and response measures should be in place at different time periods to take full advantage of limited public health resources and minimize the negative consequence of heatwaves.

Potential mechanisms for heatwaves and subsequent sudden increases in dengue cases are unclear. One possible explanation is that long-lasting high temperatures affect the mosquito population. High temperatures during heatwaves may exceed the threshold for mosquito survival and mosquito population would be greatly reduced [15]. However, the later temperature drops and high humidity in the region under study will provide opportunities for the further development of mosquito larvae or pupae, finally resulting in mosquito population abundance. This assumption is supported by a model simulation study suggesting that hot temperatures result into delayed mosquito outbreaks in Thailand [38]. It may be possible, albeit not yet been demonstrated, that after heatwaves weather conditions return to normal such as high humidity and appropriate temperature, which are favourable for the development of dengue virus (reduced extrinsic incubation period). Another reason may be related to the change in human activities during and after heatwaves; lower temperature after heatwaves will encourage people to spend more time outside, increasing the likelihood of being bitten by mosquitoes. Unfortunately, previous mechanistic research solely focused on identifying the preferable temperature

range for mosquitoes from the whole range of temperature [13], providing little information on the effects of heatwaves (sustained high temperatures). There is abundant evidence that heatwaves increase the risk of morbidity and mortality from non-infectious diseases such as cardiovascular and respiratory diseases [39]. High temperatures during heatwaves could have negative effects on human cardiovascular, respiratory and immune functions [40, 41], which likely increase the population's vulnerability to infectious diseases such as dengue. In the light of longer and more frequent heatwaves expected as a result of climate change [8], more research is required to unmask how heatwaves affect mosquito population and biting behaviours, as well as other conditions in favour of dengue transmission such as population movement and imported dengue cases.

To the best of our knowledge, this is the first study to link heatwaves to dengue outbreaks. If our findings can be confirmed in other regions of the globe, heatwaves could be used for the development of early warning system. We also demonstrated different non-linear relationships between temperature and dengue incidence during different outbreaks. Relevant results such as temperature thresholds will provide more specific guidance for the prevention and control of dengue. The dengue burden assessment will help policy makers, stakeholders, and health researchers to evaluate the cost-effectiveness of health policies and intervention measures.

Some limitations of this study need to be acknowledged. First, this study was conducted in one city, limiting the generalizability of results to other regions with different characteristics. Second, only short-term effects of temperature were considered in the assessment of temperature-dengue association. We may have underestimated temperature-related dengue burden considering the long-term effects of temperature on dengue. Thirdly, we only used weekly data and were unable to quantify the onset, duration and frequency of heatwaves on the dengue incidence. Fourthly, due to the issue of data availability, we cannot stratify our data analysis by the serotypes of dengue, which warrants further investigation in future research [42].

## Conclusions

There is a non-linear short-term relationship between temperature and dengue incidence, varying by the level of outbreaks. A large number of dengue cases can be attributed to temperature. This study also suggests that heatwaves may delay the timing of dengue outbreaks and increase the magnitude of outbreaks. If the relationship between heatwave and dengue is confirmed by other studies, these findings may facilitate the development of early warning systems for controlling and preventing dengue transmission.

## Supporting information

**S1 Fig. Model's goodness-of-fit check for small dengue outbreaks.**
(TIF)

**S2 Fig. Model's goodness-of-fit check for medium dengue outbreaks.**
(TIF)

**S3 Fig. Model's goodness-of-fit check for large dengue outbreaks.**
(TIF)

**S4 Fig. Exposure-response relationship between rainfall and risk of dengue.** Black lines indicate the relative risk and the shaded area the 95% confidence interval; dotted lines are the threshold temperature with the highest relative risk.
(TIF)

**S5 Fig. Exposure-response relationship between week temperature variation and risk of dengue.** Black lines indicate the relative risk and the shaded area the 95% confidence interval; dotted lines are the threshold temperature with the highest relative risk.
(TIF)

**S6 Fig. Exposure-response relationship between temperature and risk of dengue, using different thresholds for outbreak definition.**
(TIF)

**S7 Fig. Estimated relative risk of heatwaves on dengue incidence, using different heatwave definitions.** y-axis represents the relative risk; x-axis represents the lag weeks from 0 to 20; Three heatwave definitions were used including: (1) seven or more days with temperature exceeding 90th percentile of daily temperature distribution; (2) two or more days with temperature exceeding 99th percentile of daily temperature distribution; and (3) three or more days with temperature exceeding 99th percentile of daily temperature distribution.
(TIF)

## Acknowledgments

We would like to thank Hanoi Center for Preventive Medicine for providing the dengue data and National Oceanic and Atmospheric Administration for making climate data public available.

## Author Contributions

**Conceptualization:** Jian Cheng, Wenbiao Hu.

**Data curation:** Jian Cheng, Do Thi Thanh Toan, Pham Quang Thai, Zhiwei Xu, Wenbiao Hu.

**Formal analysis:** Jian Cheng, Zhiwei Xu, Wenbiao Hu.

**Funding acquisition:** Hilary Bambrick, Laith Yakob, Gregor Devine, Francesca D. Frentiu, Wenbiao Hu.

**Investigation:** Jian Cheng, Do Thi Thanh Toan, Pham Quang Thai, Zhiwei Xu, Wenbiao Hu.

**Methodology:** Jian Cheng, Hilary Bambrick, Laith Yakob, Gregor Devine, Francesca D. Frentiu, Zhiwei Xu, Wenbiao Hu.

**Project administration:** Zhiwei Xu, Wenbiao Hu.

**Resources:** Jian Cheng, Zhiwei Xu, Wenbiao Hu.

**Software:** Jian Cheng, Zhiwei Xu, Wenbiao Hu.

**Supervision:** Hilary Bambrick, Wenbiao Hu.

**Validation:** Jian Cheng, Hilary Bambrick, Do Thi Thanh Toan, Pham Quang Thai, Zhiwei Xu, Wenbiao Hu.

**Visualization:** Jian Cheng, Laith Yakob.

**Writing – original draft:** Jian Cheng.

**Writing – review & editing:** Jian Cheng, Hilary Bambrick, Laith Yakob, Gregor Devine, Francesca D. Frentiu, Do Thi Thanh Toan, Pham Quang Thai, Zhiwei Xu, Wenbiao Hu.

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
