## [Decision Letter · Decision Letter 0]

11 Oct 2019

Dear Mr Hu:

Thank you very much for submitting your manuscript "Heatwaves and dengue outbreaks in Hanoi, Vietnam: new evidence on early warning" (PNTD-D-19-01089) for review by PLOS Neglected Tropical Diseases. Your manuscript was fully evaluated at the editorial level and by independent peer reviewers. The reviewers appreciated the attention to an important topic but identified some aspects of the manuscript that should be improved.

We therefore ask you to modify the manuscript according to the review recommendations before we can consider your manuscript for acceptance. Your revisions should address the specific points made by each reviewer.

(1) A letter containing a detailed list of your responses to the review comments and a description of the changes you have made in the manuscript.

(2) Two versions of the manuscript: one with either highlights or tracked changes denoting where the text has been changed (uploaded as a "Revised Article with Changes Highlighted" file ); the other a clean version (uploaded as the article file).

(3) If available, a striking still image (a new image if one is available or an existing one from within your manuscript). If your manuscript is accepted for publication, this image may be featured on our website. Images should ideally be high resolution, eye-catching, single panel images; where one is available, please use 'add file' at the time of resubmission and select 'striking image' as the file type. 

Please provide a short caption, including credits, uploaded as a separate "Other" file. If your image is from someone other than yourself, please ensure that the artist has read and agreed to the terms and conditions of the Creative Commons Attribution License at http://journals.plos.org/plosntds/s/content-license (NOTE: we cannot publish copyrighted images). 

(4) Appropriate Figure Files 

Please remove all name and figure # text from your figure files upon submitting your revision. Please also take this time to check that your figures are of high resolution, which will improve both the editorial review process and help expedite your manuscript's publication should it be accepted. Please note that figures must have been originally created at 300dpi or higher. Do not manually increase the resolution of your files. For instructions on how to properly obtain high quality images, please review our Figure Guidelines, with examples at: http://journals.plos.org/plosntds/s/figures

While revising your submission, please upload your figure files to the Preflight Analysis and Conversion Engine (PACE) digital diagnostic tool, https://pacev2.apexcovantage.com/ PACE helps ensure that figures meet PLOS requirements. To use PACE, you must first register as a user. Then, login and navigate to the UPLOAD tab, where you will find detailed instructions on how to use the tool. If you encounter any issues or have any questions when using PACE, please email us at figures@plos.org.

We hope to receive your revised manuscript by Dec 10 2019 11:59PM. If you anticipate any delay in its return, we ask that you let us know the expected resubmission date by replying to this email.

To submit your revised files, please log in to https://www.editorialmanager.com/pntd/

Sincerely,

David W.C. Beasley

Associate Editor

Scott Halstead

Deputy Editor

Reviewer's Responses to Questions

**Key Review Criteria Required for Acceptance?**

**Methods**

-Are the objectives of the study clearly articulated with a clear testable hypothesis stated?

-Is the study design appropriate to address the stated objectives?

-Is the population clearly described and appropriate for the hypothesis being tested?

-Is the sample size sufficient to ensure adequate power to address the hypothesis being tested?

-Were correct statistical analysis used to support conclusions?

-Are there concerns about ethical or regulatory requirements being met?

Reviewer #1: Yes although in attached Word doc some clarification in the stats would be helpful. I am also interested to know what effect inclusion of district as random factor would have. Spatial scale seems very large but perhaps this was not the case.

Reviewer #2: The objective of the study is clearly stated although there is no testable hypothesis provided. The population has been described with mention of what is known of the disease prevalence. The sample size is sufficient though the geographical coverage is not wide enough to allow for wider applicability of the findings. There are no concerns on the ethical requirements.

**Results**

-Does the analysis presented match the analysis plan?

-Are the results clearly and completely presented?

-Are the figures (Tables, Images) of sufficient quality for clarity?

Reviewer #1: Results are good. As stated in attached Word doc, needs some work (Fig 5 I can't read) and perhaps an additional one or two results would be of general interest.

Reviewer #2: Results are clearly presented and match the analysis plan with good quality figures generated and provided.

**Conclusions**

-Are the conclusions supported by the data presented?

-Are the limitations of analysis clearly described?

-Do the authors discuss how these data can be helpful to advance our understanding of the topic under study?

-Is public health relevance addressed?

Reviewer #1: Yes, again perhaps addition of one or two limitations, especially with regard to serotype and herd immunity.

Reviewer #2: Specific comments

1. The authors state clearly how high temperatures that occur as heatwaves affect occurrence of dengue outbreaks in the following lines; “Line 350: we found that temperature was responsible for a large number of dengue cases”. Then goes on to state in line 367: Noticeably, heatwaves had a much longer delayed effect on large dengue outbreaks than small and medium outbreaks” and line 371 that These findings may suggest that not only have heatwaves delayed the timing of dengue outbreak but also increased the magnitude”

It is unclear to me (and possibly for other readers who will be interested in this work) how the delay is conceived. What would have been the expected timing of the outbreaks following a heat wave? If increased magnitude of outbreak is bad as would be expected, is increased delayed outbreak a good thing? How can this delay be explained in terms of disease incidence vis a vis rise in temperature and what are the implications to public health? 

2. In an attempt to provide biotic explanation to the model, authors have made statements that need clarification. In line 375 “long-lasting high temperatures affect the mosquito population and in line 376 “High temperatures during heatwaves may exceed the threshold for mosquito survival and mosquito population would be greatly reduced”. How would this result in increased cases or outbreaks? It sounds like it would have the opposite effect.

In real life scenario, increased temperature can affect mosquito population negatively if the relative humidity is low. However, in high humidity, mosquito survival would not be much affected, but the high temperature would increase the virus development time (extrinsic incubation period, EIP) in the mosquito hence increasing the transmission rate. This can be considered in explaining the effect of temperature on transmission hence increased outbreaks. 

3. The authors should explain how these findings have relevance to dengue outbreak prevention and response.

**Editorial and Data Presentation Modifications?**

Reviewer #1: Please see attached doc.

Reviewer #2: Revisions should entail clarification of issues raised in the conclusion above.

**Summary and General Comments**

Reviewer #1: Please see attached doc

Reviewer #2: Overall general comments; The paper is well written and addresses a re-emerging disease of growing public health concern that is greatly influenced by climatic factors. There are a few grammatical errors that the authors can address by having a third party read through for language editing. Apart from addressing the specific comments mentioned above which will clarify and improve the manuscript, I find the paper is suitable for publication in PLOS NTD.

PLOS authors have the option to publish the peer review history of their article (what does this mean?). If published, this will include your full peer review and any attached files.

Reviewer #1: Yes: Richard Paul

Reviewer #2: Yes: Rosemary Sang

---

## [Decision Letter · Decision Letter 1]

16 Dec 2019

Dear Mr Hu,

We are pleased to inform you that your manuscript, "Heatwaves and dengue outbreaks in Hanoi, Vietnam: new evidence on early warning", has been editorially accepted for publication at PLOS Neglected Tropical Diseases.

Before your manuscript can be formally accepted and sent to production you will need to complete our formatting changes, which you will receive in a follow up email. Please note: your manuscript will not be scheduled for publication until you have made the required changes. One of the reviewers has also made some recommendations regarding additional minor edits to improve the clarity of certain portions of the manuscript that we also ask you to address during final editing for publication.  

IMPORTANT NOTES

* Copyediting and Author Proofs: To ensure prompt publication, your manuscript will NOT be subject to detailed copyediting and you will NOT receive a typeset proof for review. The corresponding author will have one final opportunity to correct any errors when sent the requests mentioned above. Please review this version of your manuscript for any errors.

* If you or your institution will be preparing press materials for this manuscript, please inform our press team in advance at plosntds@plos.org. If you need to know your paper's publication date for media purposes, you must coordinate with our press team, and your manuscript will remain under a strict press embargo until the publication date and time. PLOS NTDs may choose to issue a press release for your article. If there is anything that the journal should know, please get in touch.

*Now that your manuscript has been provisionally accepted, please log into EM and update your profile. Go to http://www.editorialmanager.com/pntd, log in, and click on the "Update My Information" link at the top of the page. Please update your user information to ensure an efficient production and billing process.

*Note to LaTeX users only - Our staff will ask you to upload a TEX file in addition to the PDF before the paper can be sent to typesetting, so please carefully review our Latex Guidelines [http://www.plosntds.org/static/latexGuidelines.action] in the meantime.

Best regards,

David W.C. Beasley

Associate Editor

Scott Halstead

Deputy Editor

Reviewer's Responses to Questions

**Editorial and Data Presentation Modifications?**

Reviewer #1: I thank the authors for their careful consideration of my previous comments/suggestions. It now reads very well and is very clear.

However, the paper could do with a thorough check of the English. With no disrespect intended to the English of the main authors, perhaps some of the middling native English speakers could read this through as well. Fresh eyes to spot the errors.

For example (and there are issues throughout, albeit minor)

In the abstract

Conclusions: The short-term association between temperature and dengue risk varied by

52 the level of outbreaks and temperature seem more likely affect large outbreaks. Moreover,

53 heatwaves may delay the timing and increase the magnitude of dengue outbreaks.

Highlighted sentence doesn’t make sense. The “seem more likely affect large outbreaks” hangs on its own.

Author summary

73 such as El Niño event and extreme hot summer can affect dengue incidence rate

“event… summer ….. rate” should be plural

……. Many other examples throughout

“Finally” Figure 5 legend shows “Dealyed….”

PLOS authors have the option to publish the peer review history of their article (what does this mean?). If published, this will include your full peer review and any attached files.

Reviewer #1: Yes: Rick Paul

---

## [Editor Report · Acceptance letter]

13 Jan 2020

Dear Mr Hu,

We are delighted to inform you that your manuscript, "Heatwaves and dengue outbreaks in Hanoi, Vietnam: new evidence on early warning," has been formally accepted for publication in PLOS Neglected Tropical Diseases.

Best regards,

Serap Aksoy

Editor-in-Chief

Shaden Kamhawi

Editor-in-Chief
